# Effectiveness of unguided internet-based computer self-help platforms for eating disorders (with or without an associated app): A systematic review

**Alessandra Diana Gentile**[1]*, **Yosua Yan Kristian**[2], **Erica Cini**[1,2,3]

**1** Department of Child & Adolescent Psychiatry, Institute of Psychiatry, Psychology & Neuroscience, King's College London, London, United Kingdom, **2** Division of Medicine, University College London, London, United Kingdom, **3** East London NHS Foundation Trust, London, United Kingdom

* alessandra.d.gentile@kcl.ac.uk

## Abstract

### Background

Following the COVID-19 pandemic, internet-based computer self-help platforms for eating disorders (EDs) became increasingly prevalent as a tool to effectively prevent and treat ED symptoms and related behaviours. This systematic review explored the effectiveness of unguided internet-based computer self-help platforms for EDs.

### Methods

From inception to the 31st of May 2024, a systematic search of Ovid MEDLINE, Embase, Global Health, and APA PsycInfo was conducted. This systematic review followed the Preferred Reporting Items for Systematic Reviews and Meta-Analyses (PRISMA) guidelines. Outcome quality assessments were conducted according to the Grading of Recommendations Assessment, Development and Evaluation (GRADE).

### Results

12 RCTs, with a total of 3400 participants, were included. 2 studies explored the effectiveness as primary prevention, 7 as secondary prevention, and 3 as tertiary intervention. The gathered literature demonstrated unguided internet-based computer self-help platforms as effective in reducing ED core symptoms and related behaviours, with psychoeducation, cognitive behavioural, and dissonance-based approaches being the most prevalent approaches.

### Conclusions

Unguided internet-based computer self-help platforms are effective in the short-term reduction of ED symptoms and associated behaviours and should be implemented in the early stages of a tiered healthcare system for ED treatments.

**Data availability statement:** Supplemental data sets are provided in S1 Appendix. All findings are fully available, without restrictions.

**Funding:** The author(s) received no specific funding for this work.

**Competing interests:** The authors have declared that no competing interests exist.

## Trial registration

Prospero (CRD42024520866).

## Author Summary

With the rise of the COVID-19 pandemic, online self-help programs for eating disorders (EDs) have become more common. These programs aim to prevent and reduce ED symptoms without direct guidance from a therapist. This review examined how effective these unguided internet-based self-help platforms are for ED prevention and treatment. Researchers systematically analyzed 12 clinical studies involving 3,400 participants. The studies explored the use of these programs in different stages of ED care: some focused on preventing EDs, while others targeted individuals already experiencing symptoms. The results showed that these online platforms effectively reduced ED symptoms and harmful behaviors, especially when using educational, cognitive-behavioral, or dissonance-based methods. Overall, the findings suggest that unguided online self-help programs can be a valuable early intervention tool in ED treatment. They provide a practical, accessible, and effective way to support individuals struggling with ED symptoms, especially in the short term. Healthcare systems should consider integrating these platforms as part of a stepped-care approach to ED management.

## Introduction

### Eating disorders

Eating disorders (ED) are a group of psychiatric disorders characterised by irregular eating behaviours, body image concerns, and, in some types, fear of gaining weight [1]. EDs are underpinned by several cognitive and affective risk factors that exacerbate their development and persistence [2]. Core components include thin idealization, which refers to the societal and personal endorsement of an unrealistically slim body type as the ideal standard of beauty [3]. This often leads to body dissatisfaction—a pervasive sense of unhappiness with one's physical appearance—that is a strong predictor of ED onset [4]. Additional psychological factors such as depression, perseverative thinking (repetitive, negative thought patterns), and resistance to change also play significant roles in both the aetiology and progression of EDs [5]. These elements not only impair quality of life but also create barriers to recovery, reinforcing the urgency for targeted and accessible interventions [6]. Early identification and addressing these factors through psychoeducation and evidence-based interventions are critical in mitigating the risks associated with EDs.

Literature shows that 5.5% to 17.9% of females and 0.6% to 2.4% of males are diagnosed with EDs based on Diagnosis and Statistical Manual of Mental Disorders (DSM-5) criteria [7]. Moreover, some minority groups are at particular risk of developing EDs. For example, gender and sexual minorities are demonstrated to be at greater risk for developing EDs, with anorexia nervosa and bulimia being the most prevalent [7–10]. Notably, compared to any other psychiatric condition, anorexia nervosa has the highest suicidality and mortality rates, and lowest quality of life levels, highlighting the importance for urgency of care [11,12].

Weissman and Roselli (2017) stated that merely 25% of individuals with ED symptoms or developed EDs access care, which is explained by the lack of accessible treatment and individual treatment preferences [13]. EDs have been demonstrated to negatively impact the

psychological, cognitive, physical, and social development of individuals. This evidences the need for accessible resources which target early identification through psychoeducation, such as self-help platforms.

## ED intervention

Psychoeducation improves recognition by identifying the symptoms and warns patients about the negative impact of EDs on their physical and mental health, which in turn increases awareness and overall demystifies the disorder [14,15]. This highlights that psychoeducation is an effective tool for the prevention of EDs and promoting well-being among the public [16,17]. In line with the National Institute for Health and Care Excellence (NICE) guidelines, ED interventions follow the stepped care model [18]. The stepped care model suggests that individuals should first receive low-intensity interventions such as psychoeducation and progressively receive more intensive interventions if necessary [19]. This aligns with the THRIVE framework, which highlights a needs-led approach to delivering person-centred integrated interventions [20]. According to the THRIVE framework, significance is placed on the prevention of mental health issues and the promotion of wellbeing in the general public. Further, COVID-19 put conventional ED services under amplified pressure due to the increase in referrals and acuity [21]. This places self-help platforms in good stead to provide support to the general public and to those who require early intervention.

## Self-help platforms

Self-help platforms support individuals through early intervention and resource signposting by increasing their understanding of their symptoms [15]. Additionally, some platforms help with the development of coping skills, through for example, cognitive behavioural therapy (CBT), dissonance-based intervention (DBI), motivational-enhancement therapy (MET)-in form techniques, as well as directing people to seek early help [22–24].

There are various self-help platforms, such as web-based, internet-based computer programs, offline computer programs, and mobile applications [25–30]. Computer self-help platforms allow for accessible and scalable support, taking into consideration the unlimited demand and the limited resources [31–33]. Various digital modalities have been tried, such as compact disc read-only memory (CD-ROM), videos, and text messages, but no strong evidence emerged [34,35]. Self-help platforms can be delivered with guidance from clinicians or peers, or without any guidance [23,28,36–38]. Self-help platforms mitigate barriers such as large geographical distances from healthcare clinics, treatment-seeking stigma, and time constraints, whilst also reducing ongoing staff-related treatment costs [37,39]. Self-help platforms also have a role in creating a 'waiting well' environment. This refers to a proactive approach in which individuals and their families engage in purposeful activities, such as self-guided psychoeducation to manage and mitigate the impacts of EDs and disordered eating behaviours (DEBs) [40,41]. Overall, this ensures progress and better symptom management, reduced anxiety, and planning ahead prior to engaging with formal interventions [42].

Self-help platforms can be utilised as part of therapy using a guided approach. A guided approach involves the support of a clinician or peers while the user navigates the platform and learns accurate information. Systematic reviews which compared guided to non-guided self-help platforms effectiveness in reducing ED symptomology found that guided interventions are significantly more effective compared to nonguided self-help interventions [16,43]. Additionally, guided self-help platforms were shown to significantly increase intervention adherence and participant satisfaction [44]. On the contrary, Aardom (2017) showed that an unguided self-help platform is more suitable for patients with EDs who exhibit mild to

moderate bulimic symptoms, but less effective for those with severe symptoms [45]. This can be explained by the mild and moderate symptomologies requiring less intensive support; therefore, allowing individuals to manage their symptoms more effectively. Contrastingly, severe symptomologies require personalized and guided interventions with clinical support which addresses the complexity of symptoms. Therefore, although self-help platforms have demonstrated positive outcomes, face-to-face CBT showed quicker and better reductions in abstinence rate and ED psychopathology in adults with EDs. [28].

### Internet-based computer self-help platforms

In some cases, computer platforms are delivered online through internet-based platforms or websites, which we define as internet-based computer self-help platforms. These platforms may, at times, be accompanied by smartphone apps, which serve as supplementary tools but are not standalone interventions. Internet-based computer self-help platforms which utilise internet cognitive behavioural therapy (ICBT or CBI-I) and internet dissonance-based intervention (IDBI) have been demonstrated to be feasible and effective alternatives [46,47]. There is evidence that internet-based computer self-help platforms effectively reduce ED symptoms and ED-related behaviours [45,48].

### Aim

As evidenced, while previous systematic reviews have compared guided to non-guided self-help platforms, there is a gap in research collating findings regarding the effectiveness of only unguided internet-based computer self-help platforms for people with EDs [16,43,49]. Moreover, unguided internet-based computer self-help platforms are potentially effective in reducing risk for people with ED symptoms [47,48,50]. Therefore, this systematic review aims to evaluate the effectiveness of unguided internet-based computer self-help platforms for several outcomes: (1) global ED symptoms, (2) ED-related behaviours, such as thin idealisation, body dissatisfaction, quality of life, depression, perseverative thinking, and resistance to change, and (3) preventing the onset of EDs. This review emphasizes interventions where the unguided internet-based computer self-help platforms serve as the primary delivery method. The decision to focus on unguided internet-based computer self-help platforms stems from their unique design and functionality, which may differ from standalone mobile applications in terms of usability, accessibility, and therapeutic structure.

## Method

### Search strategy

A systematic search was conducted following the guidelines of the Preferred Reporting Items for Systematic Reviews and Meta-Analyses (PRISMA) [51]. The protocol for this systematic review has been registered on PROSPERO with registration number: CRD42042520866 and the protocol was published on JIMR journal [52]. Although Wilksch et al. 2018 was initially included in the protocol as a potentially eligible study, it was excluded from the main manuscript during the full-text screening phase because the intervention was delivered solely via a mobile app, without a corresponding website version, and therefore did not meet the study design criteria. Four databases, Ovid MEDLINE I (inception: 1946), Embase (inception: 1974), Global Health (inception: 1910), and APA PsychInfo (inception: 1806), were used to identify relevant literature from inception to 31 May 2024. The search terms are depicted in Table 1. A manual search of references was conducted utilising Google Scholar to identify alternative literature. The search was conducted by AG and YYK and reviewed by EC.

**Table 1. Example of search strategy on Ovid Medline.**

| Population | 1. Eating Disorder* |
| | 2. Anorexia |
| | 3. Anorexia nervosa |
| | 4. Bulimia |
| | 5. Binge eating |
| | 6. Avoidant restrictive food intake disorder |
| | 7. ARFID |
| | 8. Otherwise specified feeding or eating disorder |
| | 9. OSFED |
| | 10. OR/1–9 |
| **Intervention** | 11. Intervent* |
| | 12. Treatment* |
| | 13. Psychoedu* |
| | 14. OR/ 11–13 |
| **Treatment implemented** | 15. ICBT |
| | 16. Internet cognitive behavioral therapy |
| | 17. Self-help |
| | 18. Digital |
| | 19. Online |
| | 20. OR/15–20 |
| **Combination of search** | 21. 10 AND 14 AND 20 |

## Eligibility criteria (Table 2)

**Table 2. Eligibility criteria.**

| | Inclusion | Exclusion |
|---|---|---|
| **Population** | People with ED diagnosis | |
| | People with ED symptoms: People with a preoccupation with food and gaining weight, with reported purging, bingeing, restricting, or exercising behaviors | |
| | People at risk of developing EDs: People with weight concern scales ≥34 | |
| | People from the general population without ED-related behaviors | |
| **Intervention** | Unguided internet-based computer self-help platforms for EDs and ED-related behaviors | Self-help platforms for conditions other than EDs |
| | Interventions delivered primarily via internet-based computer self-help platforms or websites, regardless of whether they were accompanied by supplementary smartphone apps | Studies focusing exclusively on standalone mobile smartphone applications as the primary intervention |
| | Interventions targeting eating disorder symptoms or behaviors, delivered through structured, self-guided programs designed for personal computers | Studies that did not specify the platform or delivery method of the intervention. |
| **Comparator** | In-person treatment | |
| | Guided self-help platforms | |
| | Waitlist | |
| | Video or leaflet | |
| **Outcomes** | ED symptoms | Only reported physical outcomes |
| | ED-related behaviours: thin idealisation, body image, body shape concerns, weight concerns | |
| | Comorbidities: anxiety, depression | |

*(Continued)*

**Table 2.** (Continued)

|  | Inclusion | Exclusion |
|---|---|---|
| **Study design** | Research with original data, including grey literature | Systematic reviews |
|  | Literature in English | Meta-analyses |
|  |  | Posters |
|  |  | Leaflets |
|  |  | Books |
|  |  | Reviews |

## Screening strategy

After the search was completed, studies were exported to EndNote, and duplicates were removed. AG and YYK then initially screened articles independently by titles and abstracts. Following this, the authors screened the full text of the literature and collected research that abided by the inclusion and exclusion criteria. AG and YYK did not find any discrepancies. Any disagreements were resolved by discussion with the senior author, EC.

## Data extraction

AG and YYK independently completed data extraction. The key aspects were recorded on an Excel table based on the following variables: the primary author's name and published date; participants profile, baseline sample number, female gender percentage and mean age; follow-up times and sample size; outcome measure; intervention and comparison group; key findings; and overall risk of bias. AG and YYK inputted the data independently for each included study to assess the study's eligibility. There were no discrepancies during data extraction. EC would have been consulted for any uncertainty or unresolved disagreements.

The included studies were separated into three categories based on the participant populations and the implemented intervention. Based on the stepped and THRIVE approach, studies included in the primary prevention were participants from the general population. Studies included in the secondary prevention involved participants who were at risk of developing EDs and studies in the tertiary prevention included participants meeting the clinical ED threshold. The authors categorised thin idealisation, body image, body shape concerns, and weight concerns as ED-related behaviours.

## Quality assessment

The Grading of Recommendations Assessment, Development and Evaluation (GRADE) method was used to analyse the quality of the studies [53]. The GRADE method was performed to analyse the risk of bias, inconsistency, indirectness, imprecision, and publication bias ranging from 'no serious inconsistency' to 'very serious'. Additionally, the revised Cochrane risk-of-bias assessment (RoB 2) was utilised to assess the risk of bias of the randomised controlled trials (RCTs) using the Cochrane Review Manager software. All included literature were RCTs, and no additional research methods were detected. The five domains of potential risk-of-bias explored were randomisation, divergence from the intended intervention, analysis of missing data, measurement of outcomes, and selection of reported results. AG and YYK independently reviewed each included study and determined levels of potential risk-of-bias for each domain, ranging from 'low' to 'high'. The reporting of the study adhered to the PRISMA 2020 guidelines.

## Strategy for data synthesis

The systematic review findings were summarised in an Excel table and then narratively synthesised and analysed based on Popay et al.'s (2006) guidelines. Included studies were analysed

for clinical effectiveness and ED related behaviors, and ED prevention). Effectiveness was reported in terms of the unguided internet-based computer self-help platforms' effectiveness in decreasing ED symptoms and psychopathologies. ED related behaviors were reported in terms of the unguided internet-based computer self-help platforms' effectiveness in reducing ED related mental health difficulties (i.e., perseverative thinking, body dissatisfaction, thin idealization, fear of becoming fat, preoccupation of food and weight, motivation to change their weight, self-esteem, depression, and quality of life). ED prevention was reported in terms of the ability of the unguided internet-based computer self-help platforms' to reduce the onset of ED eating behaviors.

Most of the studies had a high amount of missing data, so the final data gathered from participants who completed the study was assessed. There were variances in the population target of the included studies, thus, the authors split the findings into three sections according to the population: primary, secondary, and tertiary prevention. Primary prevention was used for research in generally healthy populations, secondary prevention was used for research conducted with participants who were at risk of developing EDs, lastly, tertiary prevention was used for research in participants meeting the clinical threshold of EDs. These domains were established to make the effect of the treatment in different population groups comparable.

## Results

### Study selection

The review identified a total of 4759 studies. Following the removal of duplicates ($n$= 2057), 2702 studies remained. After screening of titles and abstracts, 2556 studies were ineligible based on the exclusion criteria. 146 studies were screened by full text. 134 studies were excluded based on the exclusion criteria and twelve studies were subject to narrative synthesis and included in this systematic review. Most studies were excluded due to having a guided feature and no self-help comparison group or due to the self-help intervention not being virtually delivered, but rather delivered via booklets. Fig 1 depicts the PRISMA flow-chart of the selection process. S1 Appendix shows the data search results.

### Study characteristics

The included sample size ranged from 89 to 680 participants ($n$= 3400). Participants were recruited from the general public, from individuals at risk of developing EDs, or from those who met the clinical criteria for or were diagnosed with an ED. The mean age of participants was 23 years, ranging from 15 to 50 years old. 98.8% of participants were female. All studies included were RCTs. Studies took place in four continents, Europe ($n = 6$), North America ($n$= 4), Asia ($n = 1$), and Oceania ($n$= 1). All studies utilised self-report measures and employed questionnaires which measured the clinical effectiveness and mental health outcomes of the intervention, related to ED symptoms and ED related behaviors. The characteristics of the included studies are depicted in Table 3.

### Quality assessments and Cochrane risk-of-bias analysis for included studies

The risk of bias assessment for each study is presented in Fig 2. Only one study showed an overall low risk of bias [23]. Of the 12 included RCTs, missing outcome data was the most common cause for bias, found in $n$=9 (75%) studies, with low missing data found in the rest of the studies [23,48,54]. Only two studies showed measurement bias, making it the least common cause of bias [54,55]. All studies found no risk of bias due to deviations from the intended intervention. Overall, the risk of bias among the studies was mainly categorised as

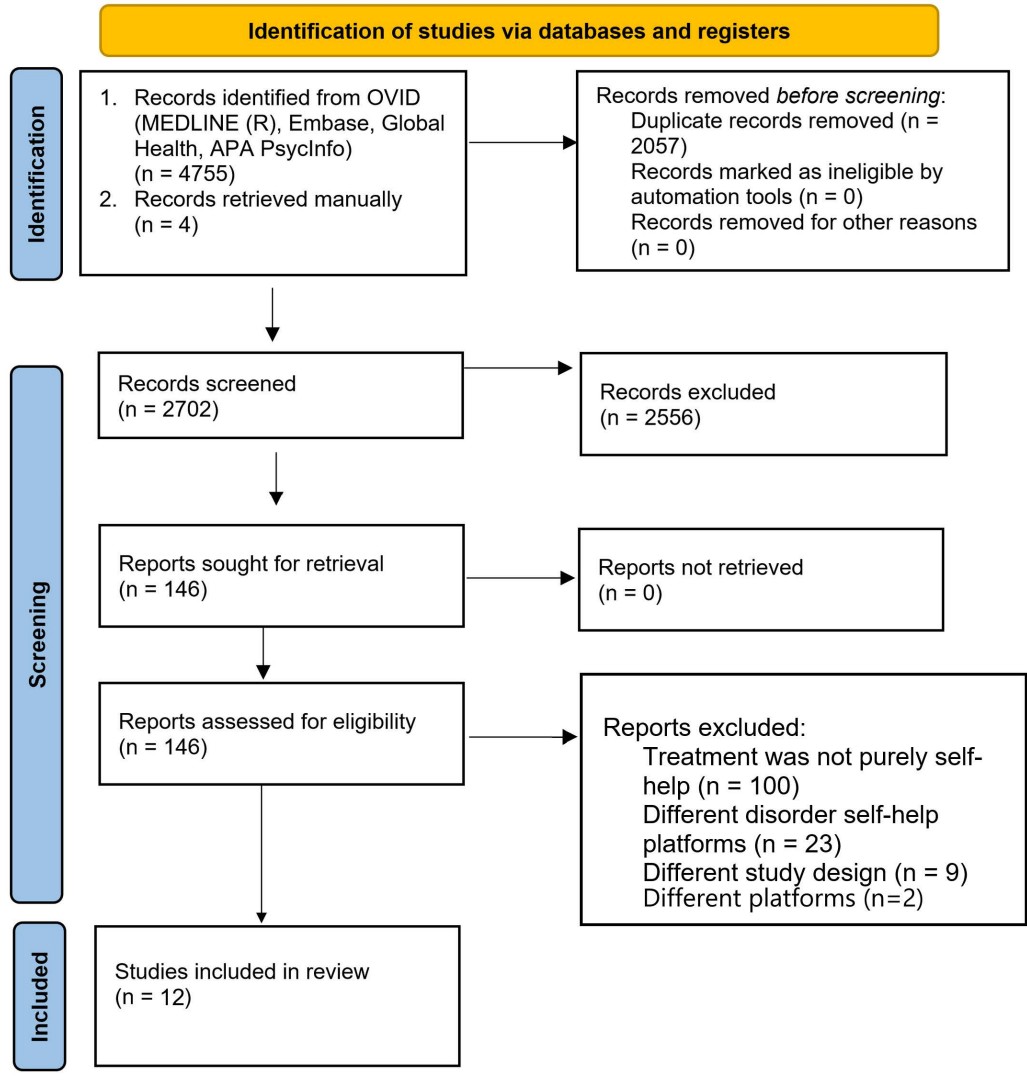

**Fig 1. PRISMA flow chart of study selection.**

some concerns. The percentage of the risk of bias is presented in Fig 3. Most studies are of moderate quality. The GRADE profile of the studies is shown in Table 4.

### Effectiveness of unguided internet-based computer self-help platforms

This systematic review included studies conducted with the general population (*n*= 2) [23,56], with participants at risk of developing EDs (*n*= 7) [46–48,54,55,57,58], and participants with ED symptoms or EDs (*n*=3) [29,30,45]

Most studies (*n*=8) utilised the EDE-Q to measure changes in ED psychopathology, which has been found to be highly sensitive for ED detection, 0.83 and a specificity of 0.96 (Cronbach's α=0.85 to 0.93, reliability r=0.68 to 0.74) [59,60]. Other ED psychopathology tools included DRES (α = 0.95, r=0.82), IBSS (α = 0.73, r=0.80), and BSQ (α = 0.96, r=0.88).

All included studies demonstrated that participants in the intervention groups demonstrated significant improvements in ED symptoms and ED-related behaviors compared to the

**Table 3. Characteristics of included studies.**

| Author(s) and date | Participant profile, Baseline sample (N) Female % and Mean age | Follow-up times and sample size (% of the initial sample) | Outcome Measure | Intervention and comparison group(s) | Key findings | Risk of bias |
|---|---|---|---|---|---|---|
| **Primary interventions** | | | | | | |
| Luo et al., 2021 | Self-reported body dissatisfaction, N=372, 100% female, Mean age 17.36. | 6-week post-treatment N=365 (98.1%) 6-month follow-up N=249 (66.9%) | Primary outcome measure: BDS Secondary outcome measure: IBSS revised, CES-D, RSES, DRES, EDDS, BAS-2 | Psychoeducation versus educational brochure control condition | Significant improvements were found for all outcome variables in the intervention group: ED symptoms F(2,740)=19.64, p<0.001, restrained eating behaviours F(2,740)=21.07, p<0.001, body dissatisfaction F(2,740)=22.80, p<0.001, thin body idealisation F(2,740)=10.77, p<0.001, depression F(2,740)=11.26, p<0.001, self-worth F(2,740)=22.45, p<0.001, and body appreciation F(2,740)=27.65, p<0.001. | Some concerns |
| Stice et al., 2020 | People who endorsed body image concerns, N= 680, 100% female, Mean age 22.2 | Post-test N=625 (91.9%), 6-month follow-up N=610 (89.7%),1-year follow-up N=613 (90.1%), 2-year follow-up N=597 (87.7%), 3-year follow-up N= 584 (85.8%), 4-year follow-up N=551 (81%). | Primary outcome measure: TIIS Secondary outcome measure: BPS, DRES, PANAS-X, EDDI | Psychoeducation versus psychoeducation with clinician or peer-leader assistance versus educational video control condition | The computer-based psychoeducation showed a significant improvement (d= -0.32, p=0.012) in thin-ideal internalisation compared to the control up to the 3-year follow-up. For body dissatisfaction, there was a significant improvement in computer-based psychoeducation (d= -0.17, p=0.044) versus control condition up to the 2-year follow-up period. No negative affect improvement was found in the computer-based psychoeducation versus control condition during any follow-up periods. Dieting improved in the computer-based psychoeducation versus control in all follow-up periods (d= -0.30, p=0.013). ED symptoms improved in the computer-based peer group and psychoeducation group (d=-0.25, p=0.007) versus the control group up to a 2-year follow-up. ED onset was significantly lower in the peer-led group versus control (HR=0.43, p=0.020, NNT=11). Computer-based psychoeducation and clinician-led did not show significant differences in ED onset compared to control. | Low |
| **Secondary Interventions** | | | | | | |
| Chithambo & Huey, 2017 | People at high risk of developing EDs with the weight concern scale score of ≥34, N= 271, 100% female, Mean age 20.8 | Post-intervention follow-up N=195 (71%) | Primary outcome measure: DRES, EAT, Secondary outcome measure: BSQ, IBSS, BDI-II | Psychoeducation, CBI-I versus DBI-I versus NI | The intervention was effective for the prevention of ED development (p<0.05). CBI-I was more effective (p= 0.001) than DBI-I (p= 0.006) at reducing ED symptoms. DBI-I had small effect sizes (d= 0.26), and CBI-I had moderate effect sizes (d= 0.53). CBI-I and DBI-I were both effective in reducing thin idealisation, body dissatisfaction, and depression (p<0.01). There was a significant moderating factor of ethnicity. DBI-I was more effective (p= 0.006) in reducing eating pathology in minorities, whereas there was no effect found in white participants (p=0.43). CBI-I was more effective in reducing depression in minorities (p= 0.001). | Some concerns |
| Haderlein & Tomiyama, 2021 | People at high risk of developing EDs with the weight concern scale score of ≥34, N= 278, 100% female, Mean age 20.5 | Post-intervention follow-up N=176 (68%) | Primary outcome measure: RED scale, BMI | Psychoeducation, CBT-I versus DBI-I versus NI | DBI-I participants demonstrated significantly reduced RED over time compared to the NI participants (z= -2.05, p= 0.045) with a moderate effect size $\beta$ = 0.31. 21% of participants in the DBI-I intervention group, 11% of CBT-I participants and 19% of the NI participants demonstrated a clinical reduction in RED scores between pre-and post-intervention, but this was not significant (p= 0.17). A significant main effect of ethnicity demonstrated lower RED scores for Latinos compared to Asian participants ($\beta$ = –0.26, p= 0.03). | Some concerns |

*(Continued)*

**Table 3.** (Continued)

| Author(s) and date | Participant profile, Baseline sample (N) Female % and Mean age | Follow-up times and sample size (% of the initial sample) | Outcome Measure | Intervention and comparison group(s) | Key findings | Risk of bias |
|---|---|---|---|---|---|---|
| Hötzel et al., 2014 | Females reporting purging, dieting, and/or exercising behaviors with a BMI between 15 and 30, N= 212, 100% female, Mean age 26.7 | Post-intervention follow-up N=125 (59%) | Primary outcome measure: EDE-Q Secondary outcome measures: SOCQ-ED, P-CED, SES, RSES | Psychoeducation, IMET versus waiting list control | Participants in the intervention condition had a reduction in their fear of becoming 'fat' ($t= -3.51$, $p= 0.001$) and a reduction in preoccupation with food and gaining weight ($t= -4.28$, $p= 0.001$). Intervention participants had increased motivation in gaining weight ($t= -3.89$, $p= 0.001$) and stopped dieting ($t = -2.48$, $p= 0.020$). Intervention participants had increased self-esteem ($t= -4.75$, $p<0.001$). There was an overall reduction in global ED with the main effect of time. | Some concerns |
| Linardon et al., 2023 | Self-reported presence of at least one binge eating episode in last 1 month, N= 392, 93% female, Mean age 28.9 | 1 month follow-up after intervention N= 252 (64%), 2-month follow-up after intervention N= 132 (33%) | Primary outcome measure: EDE-Q, Secondary outcome measure: PHQ | Psychoeducation, ICBT, NI | Participants in the intervention group had significantly lower global ED scores ($d= -0.80$, $p<0.05$). The decreased ED scores remained stable at follow-up. Participants reported other positive mental health outcomes. Participants were satisfied with the computer-based intervention. The app effectively reduced core and associated ED symptoms. 65% drop-out rate. | Some concerns |
| Karekla et al., 2022 | Females with early signs of ED or at high risk of developing ED with scores of ≥52 on the weight concern scale, N=89, 100% female, Mean age 15.3. | Post-intervention N=58 (65.9%), 1-month follow-up (treatment group only) N=25 (40%) | Primary outcome measure: EDE-Q Secondary outcome measure: EDDS, WCS, YQOL-SF, BSQ, BIAAQ, BIAQ | ACT versus wait-list control | Post-intervention, there were significant improvements in the intervention group for weight concerns ($F_{(1,54)}=56.67$, $p<0.001$) and ED behaviours ($F_{(1,55)}=6.80$, $p=0.01$), but there was no significant difference was found in the quality of life ($F_{(1,49)}=0.46$, $p>0.05$), body shape concerns ($F_{(1,49)}=3.78$, $p=0.058$), body image concerns ($F_{(1,54)}=1.86$, $p>0.05$), or physical appearance concerns ($F_{(1,49)}=0.65$, $p>0.05$). At 1-month follow-up, significant improvements were found in weight concerns ($F_{(2,23)}=9.30$, $p<0.001$), ED behaviours ($F_{(2,23)}=14.93$, $p<0.001$), body shape concerns ($F_{(2,23)}=13.38$, $p<0.001$), body image concerns ($F_{(2,23)}=10.40$, $p<0.001$), and physical appearance concerns ($F_{(2,23)}=26.64$, $p<0.001$), but not in quality of life, ($F_{(2,23)}=7.48$, $p>0.05$). | Some concerns |
| Merwin et al., 2023 | At risk of developing ED with scores of ≥52 on the weight concern scale, N=92, 100% female, Mean age 15.2 | Post-intervention N=60 (65.2%) | Primary outcome measure: EDE-Q Secondary outcome measure: WCS, BI-AAQ | ACT versus wait-list control | Post-intervention, participants in the ACT condition demonstrated significantly lower levels of weight concerns compared to the control ($d= 2.5$, $F_{(1,82)}= 33.4$, $p<0.001$). Participants with higher attendance rates demonstrated greater improvements ($B= -8.5$, $p<0.001$). Body image flexibility was found to be a potential mediator in improving weight concerns and partially explained improvements ($B= 20.9$, $p<0.001$). At one-month follow-up, ED symptoms were lower in the treatment group compared to baseline ($B= 0.56$, $p=0.029$). | Some concerns |
| Kass et al., 2014 | At high risk of developing ED with ≥ 47 on weight concern scale, N= 151, 100% female, Mean age 21 | 2-year follow-up N= 111 (74%). | Primary outcome: EDE-Q Secondary outcome measure: BDI-II, MES, BMI | ICBT self-help versus ICBT self-help + guided discussion group. | Participants in the ICBT self-help + guided discussion group had significantly lower weight and shape concerns than those who only received the ICBT self-help intervention ($SE= 2.92$; $t = 3.19$; $p = 0.002$; $d = 0.52$). There was no significant difference between baseline and post-intervention on all other measures. There was no significant correlation between time spent on the intervention and the weight and shape concern scale ($r = 0.18$; $p = 0.13$). | Low |

*(Continued)*

**Table 3.** (Continued)

| Author(s) and date | Participant profile, Baseline sample (N) Female % and Mean age | Follow-up times and sample size (% of the initial sample) | Outcome Measure | Intervention and comparison group(s) | Key findings | Risk of bias |
|---|---|---|---|---|---|---|
| **Tertiary Interventions** | | | | | | |
| Aardoom et al., 2016 | People with EDs scoring ≥ 52 on the weight concern scale or an ED symptom as assessed by the SEEDQ, N=354, 98.9% female, Mean age 24.2. | Post-intervention N= 273 (77%), 3-month follow-up N= 202 (57%), 6-month follow-up N= 118 (33%) | Primary outcome measure: EDE-Q, Secondary outcome measure: SEEDQ, ED-QOL, PHQ | Psychoeducation versus Psycho-education + low intensity therapeutic support versus Psychoeducation + high intensity therapeutic support versus waiting list control condition | The computer-based intervention, Featback, was demonstrated to effectively reduce bulimic psychopathology (d= -0.16, p<0.05). Featback did not effectively reduce anoretic psychopathology (p= 0.87). The therapist supported condition did not significantly differ in efficacy from the fully automated computer-based self-help platform. Participants who received additional support from therapists reported being more satisfied compared to the fully automated self-help platform. | Some concerns |
| Rohrbach et al., 2022 | People with self-reported ED symptoms; scoring ≥ 52 on the weight concern scale or an ED symptom as assessed by the SEEDQ N=355, 343 (96.6%) female, Mean age 27.8 | 8-week follow-up N=280 (78.8%), 3-month follow-up N=252 (70.9%), 6-month follow-up N=244 (68.7%), 9-month follow-up N=233 (65.6%), 12-month follow-up N=242 (68.1%) | Primary outcome measure: EDE-Q Secondary outcome measure: PHQ, GSES, SSL-12 | Psychoeducation versus expert-patient support versus Psychoeducation + expert-patient support versus waiting list control | The three intervention conditions significantly decreased the ED symptoms over the 8-week intervention, compared to the control condition (B=-0.15, t= -3.66, p<0.001) and were consistent at 12-month follow-up (B=0.16, t= 2.26, p=0.02). There was no significant difference in effectiveness between the intervention conditions. Participants in the intervention conditions significantly had an improvement in their depression and anxiety symptoms (B= -0.42, t= -3.85, p<0.001), and this was consistent at the 12-month follow-up (B= 0.16, t= 2.26, P= 0.02). There was no significant time-condition interaction effect for general self-efficacy and social support. Participants who received expert-patient support were more satisfied than participants who only received psychoeducation. | Some concerns |

*(Continued)*

**Table 3.** (Continued)

| Author(s) and date | Participant profile, Baseline sample (N) Female % and Mean age | Follow-up times and sample size (% of the initial sample) | Outcome Measure | Intervention and comparison group(s) | Key findings | Risk of bias |
|---|---|---|---|---|---|---|
| Pruessner et al., 2024 | Diagnosed with Binge Eating Disorder, N= 154 (96.1%) female, Mean age 35.93. | 6-week mid-treatment assessment N= 136 (88.3%), 12 weeks post-treatment N= 127 (82.4%). | Primary outcome: EDE-Q. Secondary outcome measure: WBQ, WBI, CIAQ, PHQ, GAD, RSES, DERS | Psychoeducation + ICBT versus waitlist control | Participants in the intervention group demonstrated significant decreases in binge eating episodes compared to the control condition and to baseline figures $d= -1.00$ [95% CI, $-1.30$ to $-0.70$]; P <.001). After the intervention, participants no longer met the clinically significant threshold for a binge eating disorder (MCID 3.97 episodes; reliability = 0.84). There was an interaction between treatment and time, resulting in decreased symptoms occurring in concordance with time d= $-0.79$, 95% CI, $-1.17$ to $-0.42$; P <.001. There were no significant symptom changes for participants in the control condition $d= -0.09$ [95% CI, $-0.33$ to $0.15$]; P =.44. There were significant reductions in clinical impairment for participants in the intervention group $d= -0.75$ [95% CI, $-1.13$ to $-0.37$]; P <.001), and improvements in well-being $d= 0.38$ [95% CI, $0.01- 0.75$]; P =.047. Moreover, there were decreases in depression $-0.49$ [95% CI, $-0.86$ to $-0.12$, and anxiety $d= -0.37$ [95% CI, $-0.67$ to $-0.07$]; P =.02. There were improvements in self-esteem $d= 0.36$ [95% CI, $0.13$ to $0.59$] and emotional regulation $d= -0.36$ [95% CI, $-0.65$ to $-0.07$. | Low |

BAS-2 Body Appreciation Scale-2, BDS Body Dissatisfaction Scale, BPS Body Parts Scale, CESD Center for Epidemiologic Studies Depression Scale, DRES Dutch Restrained Eating Scale, EDDS Eating Disorder Diagnostic Scale, EDDI Eating Disorder Diagnostic Interview, IBSSR Ideal Body Stereotype Scale–Revised, PANAS-X Positive Affect and Negative Affect Scale-Revised, RSES Rosenberg Self-Esteem Scale, TIIS Thin–Ideal Internalization scale, ACT Acceptance and Commitment Therapy, BDI-II Beck Depression Inventory-II, BI-AAQ Body Image Acceptance and Action Questionnaire, BIAQ Body Image Avoidance Questionnaire, BMI body mass index, BSQ-8C Body Shape Questionnaire-8C, CBI-I Internet cognitive-behavioral intervention, CBT-I Internet cognitive-behavioral treatment, DBI-I Internet dissonance-based intervention, EDE-Q Eating Disorder Examination Questionnaire, IBSS Ideal Body Stereotype Scale, ICBT Internet Cognitive Behavioural Therapy, IMET Internet motivational-enhancement therapy, MES Motivation and Expectation Scale, NI no intervention, P-CED Pros and Cons of Eating Disorders Scale, PHQ Patient Health Questionnaire, RED Reward-Based Eating Drive, RSES Rosenberg Self-Esteem Scale, SES Self-Efficacy Scale, SOCQ-ED Stages of Change Questionnaire for Eating Disorders, WCS Weight Concerns Scale, YQOL-SF Youth Quality of Life Instrument-Short Form, ED-QOL ED-Related Quality of Life Questionnaire, CIAQ Clinical Impairment Assessment Questionnaire, DERS Difficulties in Emotion Regulation Scale, GAD Generalized Anxiety Disorder Scale-7, GSES General Self-Efficacy Scale, SEEDQ Short Evaluation of Eating Disorders Questionnaire, SSL Social Support List, WBI World Health Organization Well-Being Index-5, WBQ Weekly

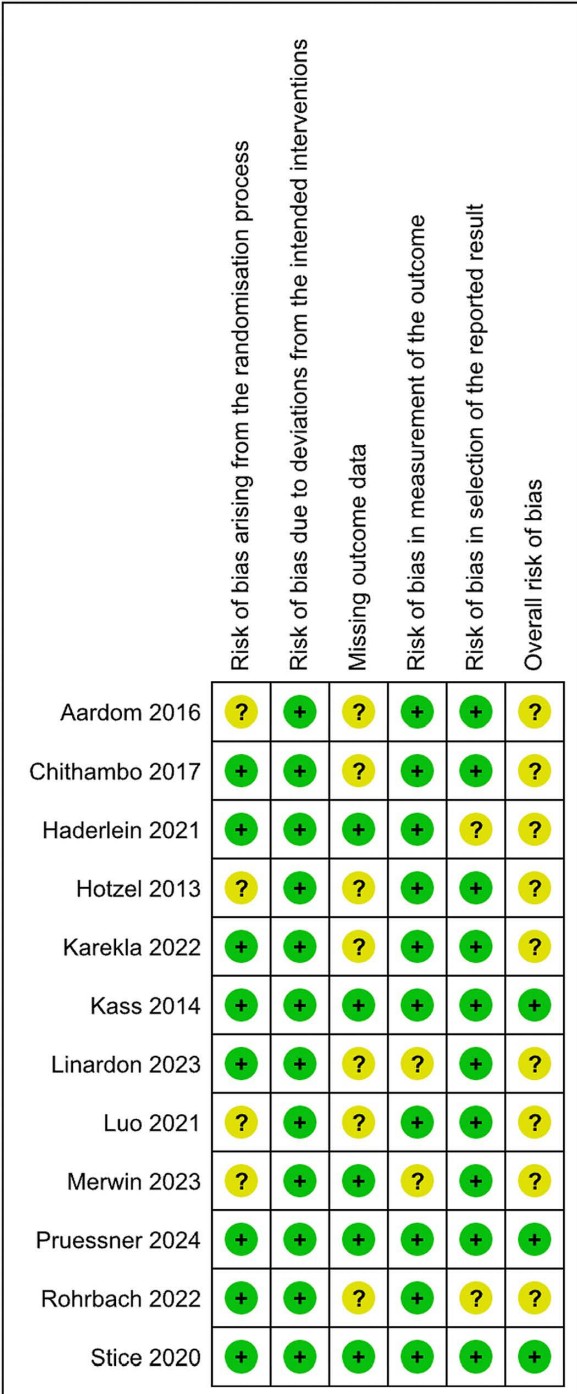

**Fig 2. Risk of bias assessment.**

control conditions. Additionally, studies that utilised unguided internet-based computer self-help platforms interventions as a primary prevention tool demonstrated them to be effective. Moreover, the studies found that the positive effects were sustained at different follow-up times, which were up to a 4-year follow-up.

**Table 4. GRADE profile of the studies.**

| Outcomes (studies) | Quality assessment | | | | | Quality Grading | Comment |
|---|---|---|---|---|---|---|---|
| | Risk of Bias | Inconsistency | Indirectness | Imprecision | Publication Bias | | |
| Eating Disorders Symptoms (*n*=11) | Serious concerns | No serious inconsistency | No serious indirectness | No serious imprecision | No serious publication bias | moderate ⊕⊕⊕◯ | - High drop-out rates and no blinding in most studies<br>- Studies used different interventions |
| ED-related behaviours: thin idealisation, body image, body shape concerns and weight concerns (*n*=6) | Serious concerns | No serious inconsistency | No serious indirectness | No serious imprecision | No serious publication bias | moderate ⊕⊕⊕◯ | - High drop-out rates and no blinding in most studies<br>- Studies used different interventions |
| Comorbidities* (*n*=2) | Serious concerns | No serious inconsistency | No serious indirectness | No serious imprecision | No serious publication bias | moderate ⊕⊕⊕◯ | - High drop-out rates and no blinding in most studies<br>- Studies used different interventions |

*Anxiety, depression

## Comparing modalities

Most studies (*n*= 11) utilised an element of psychoeducation within their intervention. On top of psychoeducation, the studies implemented ICBT techniques to address cognitive and behavioural changes in participants. The included studies which utilised psychoeducation in combination with ICBT demonstrated significantly better effectiveness compared to psychoeducation alone and psychoeducation with other modalities. Moreover, acceptance and commitment therapy (ACT) and motivational enhancement therapy (MET) were also found to reduce ED psychopathologies effectively.

Two studies compared ICBT to IDBI combined with psychoeducation. The findings are contradictory, as one of the studies suggests that ICBT is more effective than IDBI in reducing global ED psychopathologies [47]. However, the alternative study demonstrated that IDBI was more effective than ICBT in reducing reward-eating behaviours [48]. The findings demonstrated that ICBT was overall more effective compared to IDBI; however, there was a main effect of ethnicity. The results show that ethnicity has a significant impact on the effectiveness of the intervention. IDBI was demonstrated to be more effective in minority participants compared to white participants. The main effect of ethnicity suggests that different interventions may be more effective for the global majority groups versus white participants. Table 5 provides an overview of the comparison between the modalities.

## Guided versus unguided modalities

Some included studies (*n*=4) included a guided group alongside the unguided modality [23,30,45,46]. While unguided modalities showed significant improvements compared to control conditions, participants in guided interventions experienced even greater reductions in symptomatology. Additionally, participants in the guided group conditions were significantly more satisfied than participants in the unguided conditions [30,45]. Lastly, Stice et al. (2020) demonstrated significantly lower ED onset rates for participants in the peer-led groups compared to unguided and clinician-guided psychoeducation conditions.

## ED-related behaviours

Alongside global ED psychopathology, all studies measured changes in ED-related behaviours and comorbidities. All questionnaires used had good validity, with reliability scores of YQOL-SF (α = 0.80), BDI-II (α = 0.90), TIIS (α = 0.75, r=0.56), and BDS (α = 0.94,

**Table 5. Comparison between modalities.**

| Author | Platform & Aim | Structure | Main topics |
|---|---|---|---|
| 1. Luo et al., 2021<br>2. Stice et al., 2020 | eBody Project<br>(internet-based computer program)<br>Aim: Induce dissonance regarding pursuing the thin-ideal | 6 modules<br>(with homework) | Thin idealisation, discouraging thin-ideal, self-affirmation and behavioural challenge exercises. |
| Kass et al., 2014 | Student Bodies<br>(internet-based computer program and mobile application)<br>Aim: Improve dieting, excessive exercise, weight and shape concerns, binge eating, thin-body ideal internalisation | 9 modules | Discussing body image, self-esteem, effects of culture, challenging negative thoughts, and maintaining a supportive network. Coping mechanism exercises around journaling, monitoring exercise and eating habits, and goal setting for the future. |
| 1. Chithambo & Huey, 2017<br>2. Haderlein & Tomiyama, 2021 | DBI-I, contents derived from Body Project.<br>(internet-based computer program)<br>Aim: Induce dissonance regarding pursuing the thin-ideal | 4 sessions<br>(with homework) | Discussion around the effects of media on our cognition, thin idealisation, and challenging media-propagated messages of stereotypes. |
| | CBI-I, contents derived from The Body Image Workbook.<br>(internet-based computer program)<br>Aim: Reduce body image thoughts and assumptions to improve disturbed body evaluation and maladaptive eating behaviours. | 4 sessions<br>(with homework) | Psychoeducation of body image, signs and symptoms, and low mood. Discussion around how cognitive distortions maintain poor body image and negative cognitions. Managing hypothetical insensitive comments from others. |
| Hötzel et al., 2014 | ESS-KIMO<br>(web-based)<br>Aim: Enhancing motivation to change based on the transtheoretical model of change | 6 sessions | Psychoeducation regarding the transtheoretical model, ambivalence concerning change, and consequences of disordered eating. Discussion around self-esteem and future goals. |
| 1. Karekla et al., 2022<br>2. Merwin et al., 2023 | AcceptME<br>(web-based)<br>Aim: Develop skills to cope with distressing thoughts and feelings | 6 sessions | Psychoeducation regarding the management of distressing feelings and thoughts, and some strategies and coping mechanisms that can be used.<br>Recognising feelings and acceptance and practising mindfulness. Importance of supportive environment and relationships. |
| Linardon et al., 2023 | Break Binge Eating<br>(web-based and internet-based mobile application)<br>Aim: address dietary restraint, mood dysregulation, and body image concerns. | 4 modules<br>(with homework) | Psychoeducation regarding dietary restraint, mood dysregulation, and body image. |
| | Breaking the Diet Cycle<br>(web-based and internet-based mobile application)<br>Aim: address dietary restraint | 4 modules<br>(with homework) | Psychoeducation regarding regular eating and exposure to feared foods. |
| 1. Aardoom et al., 2016<br>2. Rohrbach et al., 2022 | Featback<br>(web-based)<br>Aim: educate participants about EDs and stimulate recognition and acknowledgement | Website, 4 weekly follow up | The website contains general and comprehensive information regarding body dissatisfaction, body and weight concerns, unbalanced nutrition and dieting, and compensatory behaviours. |
| Pruessner et al., 2024 | Selfapy<br>(web-based)<br>Aim: elucidate risk factors and mechanisms of BED based on a diathesis-stress model | 6 core modules and 6 optional contents | Psychoeducation regarding eating behaviour, negative thoughts, emotional regulation, stress management, and self-esteem. Coping mechanisms for mindfulness, relapse prevention and weight control. |

r=0.90). DBI-I had a smaller effect size (d=0.26) than CBI-I (d=0.53) in reducing ED-related behaviours. The findings demonstrated significant improvements in ED-related behaviours post-intervention compared to the control condition or baseline levels. Furthermore, ICBT was effective in improving self-esteem (d=0.36) and emotional regulation (d=-0.36) in patients with EDs. The findings demonstrated the improvements to be sustained at 1-month follow-up. Contrastingly, one study which utilised ACT found no significant improvement in ED-related behaviours. Interestingly, while ICBT and IDBI were both found to be effective in reducing depressive symptoms, ICBT was found to be more effective in minority groups compared to white participants [47].

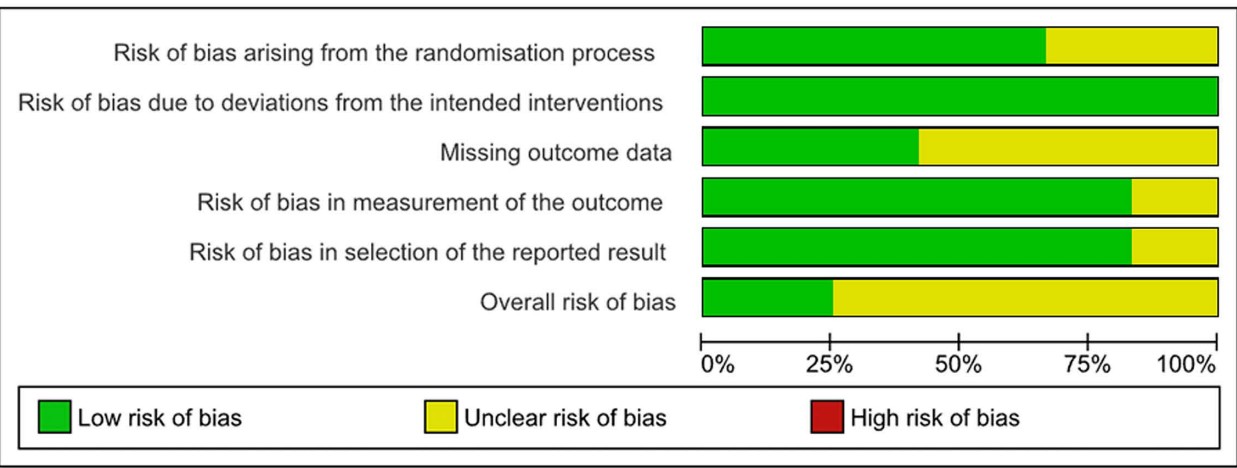

**Fig 3. Percentage of the risk of bias assessment.**

## Discussion

This systematic review found that unguided internet-based computer self-help platforms are effective in reducing global ED behaviours and ED-related behaviours. Moreover, unguided internet-based computer self-help platforms for EDs are perceived positively by users as they reduce the barriers to getting support. In line with the THRIVE framework, the self-help platforms were most effective when they were utilised as preventative measures [20]. Unguided internet-based computer self-help platforms have the potential to give the public access to evidence-based psychoeducational tools and resources [20]. This systematic review collates the clinical effectiveness of a range of unguided internet-based computer self-help platforms, despite different outcome measures being used. Unguided internet-based computer self-help platforms show a promising avenue for the management of ED symptoms, particularly when looking at a 6-month follow-up time.

Traditionally, especially pre-COVID-19, many therapeutic interventions were delivered face-to-face. For EDs, face-to-face family-based treatment for EDs and CBT-ED were the most commonly utilised therapeutic interventions supported by numerous evidence-based clinical effectiveness studies [36,50,61,62]. After COVID-19, interventions that were previously delivered face-to-face transitioned to internet-based computer self-help platforms to educate and provide resources aimed at effectively preventing and treating ED symptoms and related behaviours. While conducting this systematic review, the authors found that unguided internet-based computer self-help platforms follow similar principles to the traditional face-to-face principles. Most unguided internet-based computer self-help platforms are purely psychoeducational, CBT format (ICBT), or DBT format (IDBI). While face-to-face interventions require an in-person meeting between clinicians and patients, internet-based computer self-help platforms can be completed without assistance and overcome these barriers while maintaining the treatment effectiveness.

As shown by the results in this systematic review, of these unguided internet-based computer self-help platforms, 7 platforms were CBT-based, 2 of which looked at primary prevention, 4 looked at secondary prevention, and 1 looked at tertiary prevention. Like face-to-face CBT, ICBT platforms showed positive clinical effectiveness, particularly in patients with BED and BN. When evaluating these results, it's important to note that various outcome

measures were employed to assess clinical effectiveness, including the EDE-Q, DRES, EAT, IBSS, and BDI-II, all of which demonstrated promising findings. Across the board, 5 findings showed reductions in global EDE-Q scores; 2 findings showed a reduction in abnormal eating behaviours as measured by RED, DRES, and EAT; 2 findings showed a reduction in distorted body image and pursuit of thinness as measured by BSQ and IBSS; and 2 showed a reduction in depression, measured by BDI-II. The findings of this systematic review align with the wider literature.

## Unguided internet-based computer self-help platforms and face-to-face interventions comparison

When compared to face-to-face CBT, the findings are similar. A meta-analysis containing 79 studies [50] demonstrated that therapist-led CBT reduced short-term remission and binge or purge frequency in BN and BED compared to waitlisted conditions. A meta-analysis of 16 studies exploring ICBT effectiveness found that ICBT was effective in preventing ED in at-risk patients (-0.31 [95%CI: -0.57, -0.06] to -0.47 [95%CI: -0.82, -0.11]) and treating (-0.30 [95%CI:-0.57, -0.03] to -1.11 [95%CI: -1.47, -0.75]) AN, BN, and BED [62]. Further, binge reduction -0.66 [95%CI: -1.11, 0.22]) was also found in BED and BN patients [62]. While ICBT approaches were effective for up to 12 months, face-to-face CBT showed significant longer-term improvement (>12 months), especially in binge or purge frequency in BN (g=0.81, [95% CI 0.42 to 1.19], $p<0.01$) and BED (g=4.11, [95% CI 2.89 to 5.33], $p<0.001$) [50]. This suggests that while ICBT can be a valuable short-term treatment option, incorporating face-to-face therapy may enhance long-term outcomes and sustainability of recovery for individuals with EDs.

Another therapeutic modality commonly reported in EDs is IDBI. IDBI was the alternative self-help intervention used in comparison to ICBT. Face-to-face DBI demonstrates that these interventions are clinically significant, as shown by a meta-analysis containing 56 studies [36]. Dissonance-based prevention programs effectively reduce the thin ideal internalisation ($d = 0.57$), body dissatisfaction ($d = 0.42$), dieting ($d = 0.37$), negative affect ($d = 0.29$), and ED symptoms ($d = 0.31$) [36]. Similar results were found in this systematic review, with computer-based self-help IDBI being effective in reducing thin idealisation ($p<0.01$), body dissatisfaction ($p<0.01$), depression ($p<0.01$), and reward-based eating drive ($p=0.045$).

Studies included in this systematic review demonstrated equal outcome improvements for primary, secondary, and tertiary prevention compared to face-to-face interventions in all domains except for remission. One meta-analysis found that remission was present in 35.8% of face-to-face participants compared to 24.7% of participants in the computer-based group (RR=0.69, [95% CI 0.53 to 0.89], $p=0.004$, 4 RCTs, $n=526$) [63]. This indicates that unguided internet-based computer self-help platforms, can serve as effective early short-term preventative measures for patients awaiting face-to-face intervention. This approach not only reduces symptoms in all preventative tiers but also promotes 'waiting well,' providing valuable support for patients until they can meet with a clinician.

Linardon (2020) found that the majority of participants continue to prefer face-to-face treatment, which is highlighted in this systematic review by high drop-out rates. However, 50–70% of participants showed a willingness to use internet-based computer self-help platforms for current ED symptoms, highlighting the importance of a complementary approach. Motivation and reassurance from peers and clinicians play a crucial role in improving outcomes and reducing dropout rates for those using internet-based computer self-help platforms [16,23,26,30,46]. Factors such as current treatment experiences (b=1.18 (SE=0.26), OR=3.24, [95% CI 1.94 to 5.42]), attitude to internet interventions (b=1.97 (SE 0.20),

OR=7.15, [95% CI 4.84 to 10.58]), and stigma (b=0.47 (SE 0.14), OR=1.59, [95% CI 1.22 to 2.08]) were demonstrated as significant contributing factors to the continuation of using the unguided internet-based computer self-help platforms [64].

## Guided and unguided internet-based computer self-help platforms comparation

Whilst internet-based computer self-help platforms appear effective with regards to the prevention of EDs, it was noted that there is a significant value in the therapeutic intervention, as evidenced by the greater effectiveness of guided versus non-guided self-help platforms. Literature continues to demonstrate the importance of human interaction in symptom reduction, with included studies demonstrating greater effectiveness for guided self-help tools compared to non-guided [23,46]. Further, one study demonstrated higher participant satisfaction and more significant ED symptom reduction was found in the guided internet-based computer self-help platform [23]. Features identified in this systematic review, including engaging with peers, receiving guides from practitioners, and getting reminders improved patients' experiences during treatment [65]. However, our systematic review does not focus on guided vs unguided internet-based computer self-help platforms, therefore we suggest to review other systematic review in this focus area [66].

## Comparison between unguided internet-based computer self-help platforms

This systematic review included various unguided internet-based computer self-help platforms that employ different approaches, all of which effectively target and reduce eating disorder symptoms and related behaviours [23,30,47,55]. The studies reporting on outcomes indicated significant improvements. ICBT which focused on media internalisation was demonstrated more effective in reducing global ED pathology and improving quality of life compared to other approaches [26]. Aligned with this systematic review, the wider literature found that CBT-based short-term interventions improved the quality of life of participants with problematic social media use [67]. This can be explained in terms of the use of media focused on thinness culture and body dissatisfaction being demonstrated as a risk factor for ED development [68,69]. Moreover, ICBT provides information which educates and aims to change maladaptive behaviours, which therefore targets ED symptoms and related behaviours [26]. With the increased social media use, media-embedded clinical practice and education are needed to ensure a limited impact on media-related risk factors such as the drive for thinness and body dissatisfaction [68]. While ICBT is most effective in managing maladaptive behaviours in all groups, IDBI is most effective in reducing eating pathology, especially in the global majority groups compared to white participants [47,48].

IDBI is found to be most effective when managing specific behaviours, such as media internalisation; however, it does not target other psychopathologies, such as anorectic behaviours. Further, IDBI showed a higher effect size in thin-ideal internalisation compared to other outcomes, such as dieting, negative affect, and body dissatisfaction. These IDBI outcomes had smaller effect sizes than ICBT, suggesting that while IDBI may still provide some benefits, ICBT is generally more effective in achieving meaningful improvements in treatment outcomes.

## ED-related behaviors and comorbidities

This systematic review also found that unguided internet-based computer self-help platforms utilizing psychoeducation, ICBT, and IDBI, also improved depression and negative affect.

Face-to-face CBT that focuses on negative affect has been used to improve body shape and weight concerns in BN [70]. This approach helps individuals develop healthier coping strategies, enhance emotional regulation, and challenge distorted beliefs about body image, ultimately leading to more positive self-perceptions and reduced disordered eating behaviours. Depression and negative affect are associated with the development of maladaptive eating behaviours, which could develop into EDs [71]. Negative affect, alongside body dissatisfaction and thin-ideal internalisation, was also shown as a potential risk factor for disordered eating particularly in the Asian population, which could be explained by differences in body image culture [72].

The included samples included a significant number of participants if Asian descent, which could therefore explain the treatment effectiveness discrepancies [47,48]. A research showed people of Asian descent have lower body satisfaction compared to those from European descent, a greater desire for thinness, a more significant concern with weight, and dissatisfaction with certain body parts [73]. In addition, women from Japan have a higher body dissatisfaction compared to any other East Asian countries, where Western media was one of the factors influencing their ideal body size [69]. It is therefore, possible that Asian participants experienced more psychological discomfort while being urged to question thin ideals, which resulted in superior intervention responses. Therefore, future studies and clinical practice should account for the impact of ethnicities on treatment effectiveness.

### Drop-out rates

Despite the effectiveness, the results must be cautiously interpreted, due to the risk of bias concerns found in the studies. Most included studies have a follow-up period of 12 months or less and have a high drop-out rate, with a mean drop-out rate of 39.4%. The high drop-out rates resulted in concerns regarding the risk of bias, which was reflected in the Cochrane RoB 2 judgment. Additionally, the high drop-out rates may be attributed to the young participant sample sizes, as younger individuals tend to have a greater likelihood of disengaging. This can be explained by factors such as lack of motivation, time constraints, educational commitments, novelty seeking, previous treatment experiences, and a preference for different treatment formats [16,45,74]. Others argued drop-out rates were caused by unmet participants expectancy (OR=0.91, [95% CI 0.82 to 0.97]) and lower body mass index (OR=1.10, [95% CI 1.03 to 1.18]) [74]. These findings suggest that when participants' anticipations regarding treatment outcomes are not fulfilled, or when they have a lower body mass index, they may be more likely to disengage from the program, highlighting the importance of managing expectations and addressing individual needs in treatment.

Two studies showed a drop-out rate of less than 20% [23,29]. YYK corresponded with Stice in April 2024, discussing the retention methods. Stice et al. used several methods to improve the retention rate, including participant incentives, newsletters, and frequent follow-ups. Future researchers and clinicians should consider utilising short 5-minute telephone conversations to encourage continuing using the program which significantly improved participant adherence (T=-3.015, df =124, *p*=0.003) [75].

### Implications

This systematic review highlights the clinical effectiveness of unguided internet-based computer self-help platforms in the tiered healthcare system. Although the effectiveness varies, this study reports that the participants, the majority of whom were female, represented a diverse range of ethnicities, including individuals of American, European, Oceanian, and Asian descent. This diversity enhances the generalizability of the findings and underscores the

need for culturally tailored interventions that address the unique experiences and challenges faced by different groups.

However, as most studies demonstrate effectiveness over short follow-up periods, additional research should be conducted to evaluate the effectiveness over a longer period with decreased drop-out rates. Furthermore, while unguided self-help platforms may demonstrate short-term effectiveness, they are not recommended as substitutes for in-person treatment. However, they can play a valuable role in specific situations, particularly as a primary intervention, and their scalability makes them a convenient option in many cases. Additionally, internet-based computer self-help platforms serve as a valuable supportive tool for individuals awaiting in-person treatments, in a 'waiting well' approach.

## Future considerations

Further studies need to be conducted to explore the effectiveness of the unguided internet-based computer self-help platforms in other populations, such as the male populations and across different ethnicities, in different ED types, such as in anorexia nervosa and avoidant/restrictive food intake disorder; discussing additional ED-related behaviours, such as perfectionism, and guilt. As the attrition rates were higher in the younger populations, it is recommended that additional research is conducted to evaluate manners that retain younger users. Additionally, it would be helpful to have longitudinal studies which examine effectiveness of these tools.

Moreover, the protocol of this systematic review included two studies conducted by Wilksch et al. (2018) [26,76]. However, due to the intervention utilizing a standalone mobile app, it was ultimately excluded from this manuscript. The decision to exclude studies focusing solely on standalone mobile smartphone applications was made to provide a distinct analysis of unguided internet-based computer self-help platforms and their unique contribution to eating disorder interventions. It is acknowledged that smartphone apps represent a rapidly growing area of research in this field. Mobile app self-help platforms provide unique advantages primarily surrounding commodity and accessibility which may have serious implications for the future of unguided internet-based computer self-help platforms which are not accompanied by a mobile app version. Future research should compare mobile and computer self-help platforms for EDs to better understand the respective strengths, limitations, and potential synergies for ED treatment.

## Strengths

This systematic review has several strengths. Firstly, this systematic review found the effectiveness of different unguided internet-based computer self-help platforms as a potential temporary treatment for EDs. Secondly, the grouping of primary, secondary, and tertiary prevention clearly demonstrates its efficacy at different stages of prevention. Thirdly, the overall female-to-male ratio is representative of the prevalence of EDs worldwide. Lastly, this systematic review did not limit participant age, which improves the scope of interventions.

## Limitations

Despite the numerous strengths, this systematic review has some notable limitations. Firstly, most studies had significant drop-out rates, which could have influenced the outcome interpretations. Secondly, although most studies had good sample sizes, future studies should aim to include greater sample sizes to improve reliability and validity of the findings. Thirdly, some studies had missing data, and although personal communications with several authors had been made, no responses were given, except for the study by Stice (2020).

## Conclusion

Overall, unguided primary, secondary, and tertiary prevention internet-based computer self-help platforms are effective in reducing weight and body concerns, thin idealization, binge eating, global ED pathology, and depression, which is sustained over a 12-month period. IDBI and ICBT were the most commonly used approaches and demonstrated the greatest effectiveness. This systematic review sustains that unguided self-help platforms can be utilised to prevent the onset and the worsening of ED symptoms. However, the result needs to be interpreted cautiously, as the adherence to the intervention was low.

## Supporting information

**S1 Appendix. Data search result.**
(PDF)

## Author contributions

**Conceptualization:** Erica Cini.

**Data curation:** Alessandra Diana Gentile, Yosua Yan Kristian.

**Formal analysis:** Alessandra Diana Gentile, Yosua Yan Kristian.

**Methodology:** Alessandra Diana Gentile.

**Supervision:** Erica Cini.

**Validation:** Yosua Yan Kristian, Erica Cini.

**Writing – original draft:** Alessandra Diana Gentile.

**Writing – review & editing:** Alessandra Diana Gentile, Yosua Yan Kristian, Erica Cini.

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
