## [Decision Letter · Decision Letter 0]

16 Dec 2024

PDIG-D-24-00478Clinical Effectiveness of Computer-Based Psychoeducational Self-Help Platforms for Eating Disorders (with or without an associated App): A Systematic ReviewPLOS Digital Health Dear Dr. Gentile, Thank you for submitting your manuscript to PLOS Digital Health. After careful consideration, we feel that it has merit but does not fully meet PLOS Digital Health's publication criteria as it currently stands. Therefore, we invite you to submit a revised version of the manuscript that addresses the points raised during the review process. Please submit your revised manuscript within 60 days Feb 14 2025 11:59PM. If you will need more time than this to complete your revisions, please reply to this message or contact the journal office at digitalhealth@plos.org. Please include the following items when submitting your revised manuscript:* A rebuttal letter that responds to each point raised by the editor and reviewer(s). You should upload this letter as a separate file labeled 'Response to Reviewers '. This file does not need to include responses to any formatting updates and technical items listed in the 'Journal Requirements' section below.* A marked-up copy of your manuscript that highlights changes made to the original version. You should upload this as a separate file labeled 'Revised Manuscript with Track Changes '.* An unmarked version of your revised paper without tracked changes. You should upload this as a separate file labeled 'Manuscript '. If you would like to make changes to your financial disclosure, competing interests statement, or data availability statement, please make these updates within the submission form at the time of resubmission. Guidelines for resubmitting your figure files are available below the reviewer comments at the end of this letter. We look forward to receiving your revised manuscript. Kind regards, Calvin Or, PhDSection EditorPLOS Digital Health Calvin OrSection EditorPLOS Digital Health Leo Anthony CeliEditor-in-ChiefPLOS Digital Healthorcid.org/0000-0001-6712-6626 **Journal Requirements:**

1. We ask that a manuscript source file is provided at Revision. Please upload your manuscript file as a .doc, .docx, .rtf or .tex.

2. Please provide separate figure files in .tif or .eps format.

3. In the online submission form, you indicated that "The results of the datasets used in this systematic review are available from the corresponding author upon reasonable request.". 

a. In a public repository, 

b. Within the manuscript itself, or 

c. Uploaded as supplementary information.

4. We have amended your Competing Interest statement to comply with journal style. We kindly ask that you double check the statement and let us know if anything is incorrect. 

5. As required by our policy on Data Availability, please ensure your manuscript or supplementary information includes the following: 

**Additional Editor Comments (if provided):** It is a bit unclear what is meant by “computer-based platforms.” Are these mobile apps? If so, when the authors say “Computer-based self-help platforms ± associated apps,” it would just mean “app platforms + associated apps,” which is redundant. If these are not solely mobile apps, what other types of computers are being discussed? Desktop computers, tablet computers, or something else?

Regarding the first comment above, it is not merely about the use of terminology, but it also impacts the screening and selection of studies to be included in the review. Specifically, if the term "computer-based platforms" was not clearly defined from the beginning, it would have affected the screening and selection process conducted by the researchers.

In Table 2, for each study, please provide information about what computer technology was involved. I believe this information can be presented in the column titled “Intervention and comparison group(s).” However, if it is not appropriate there, the authors can include a new column for it. For example, in Luo et al., 2021, the intervention is described as “Psychoeducation,” but no information is provided about the technology used to deliver the psychoeducation. Please include such information for all studies in Table 2.**Reviewers' Comments:** Reviewer's Responses to Questions

**Comments to the Author**

1. Does this manuscript meet PLOS Digital Health’s publication criteria ? Is the manuscript technically sound, and do the data support the conclusions? The manuscript must describe methodologically and ethically rigorous research with conclusions that are appropriately drawn based on the data presented.

Reviewer #1: Yes

Reviewer #2: Yes

2. Has the statistical analysis been performed appropriately and rigorously?

Reviewer #1: Yes

Reviewer #2: Yes

3. Have the authors made all data underlying the findings in their manuscript fully available (please refer to the Data Availability Statement at the start of the manuscript PDF file)?

Reviewer #1: Yes

Reviewer #2: Yes

4. Is the manuscript presented in an intelligible fashion and written in standard English?

Reviewer #1: Yes

Reviewer #2: Yes

5. Review Comments to the Author

Reviewer #1: - Did you exclude studies on mobile smartphone apps? If so, there needs to be rationale for this. Additionally, computer-based self-help platforms +/- associated apps are mentioned, but does this mean smartphone apps? Please clarify in the manuscript. This implies that some studies include a computer-based platform as well as an app, but there isn’t stratification or discussion around the difference between computer-based platforms alone vs with apps. Smartphone apps do need to be addressed here, there are many studies on smartphone apps for eating disorders (Tregarthen J, Kim JP, Sadeh-Sharvit S, Neri E, Welch H, Lock J. Comparing a tailored self-help mobile app with a standard self-monitoring app for the treatment of eating disorder symptoms: Randomized controlled trial. JMIR Mental Health. 2019 Nov 21;6(11):e14972, Juarascio AS, Manasse SM, Goldstein SP, Forman EM, Butryn ML. Review of smartphone applications for the treatment of eating disorders. European Eating Disorders Review. 2015 Jan;23(1):1-1.)

- The sentences in lines 124-126 belong in the methods section

- Is there a reference for the protocol that was “published on JIMR journal” (line 132)?

- Searching databases from inception onwards was mentioned in the abstract, but not in the search strategy section of the methods. Please add this to the methods section, and consider also putting in brackets behind each database the year of inception.

- Consider also including a full search from one database as a supplementary file so readers can see exactly what terms and subject headings you used.

- Consider using PICOS (population, intervention, comparator/control, outcome, study design) to organize rows on your eligibility criteria table, it was slightly difficult to follow.

- You can remove this sentence (line 147-148): “The systematic review excluded literature that did not meet the inclusion criteria.” It is implied that studies need to meet inclusion criteria.

- Please change the wording the reason for excluding reports so that it’s consistent with your exclusion criteria. I’m not sure what “Not appropriate reporting style” means. Does this refer to publication type or study design? Or something else?

Reviewer #2: The paper presented a systematic review of the effectiveness of computer-based psycho-educational self-help platforms for reading disorder behaviors. The review is methodologically sound and the findings are an important addition to the literature regarding flexible intervention delivery across a variety of mental health concerns. The paper however has a number of areas of concern which I outline below.

1. The title "Clinical Effectiveness of Computer-Based Psychoeducational Self-Help Platforms for

Eating Disorders (with or without an associated App): A Systematic Review" is misleading. The population included are not all clinical in nature. As noted, the studies span primary, secondary, and tertiary interventions thereby including sub-clinical populations. The authors appear interested not only in whether an individual meets criteria for an ED, but also premorbid indicators.

2. Regarding the title and the paper overall, the use of the term "computer-based' is inaccurate in my view. Computer based could mean with or without the use of the internet. It is my understand that all of these interventions are internet, online, or web-based. Therefore those terms are more accurate. Computer-based could mean an external program downloaded to a computer and used offline, or in earlier tech the use of a CD-Rom program. It is important in describing these programs to use the most accurate terminology to avoid confusion.

3. Again referring to the title and the paper throughout, I found it confusing that the authors continually used the plus-minus sign to with "associated apps". The plus-minus symbol was distracting to me throughout the paper. I recommend again using a term such as web-based or online or Internet based.

4. I also believe it would be important to describe which of these interventions were browser based only, app based only, or amiable on both? Perhaps there are different drop out rates when a program is also available via an app.

5. It also seems the authors interchangeably use "self-help programs" along with their use of "computer based programs" to refer to online programs, but this is to be inferred by the reader. Again this creates confusion. A 'self help program' is not by definition computer based or internet based. Please use care and clarity in writing.

6. I found the introduction literature review to be a bit disorganize with key information combined without a clear structure or guiding framework. For example, self-help programming's ability to overcome barriers is included in the same paragraph as the impact of COVID on use of interventions as well as description of the 'waiting well'. I recommend re-working the introduction focusing on flow and organization and improved used of headings and subheadings.

7. While space may not allow for a thorough description, I think it would be helpful to provide more information and research supporting the key outcomes measured - thin idealisation, body dissatisfaction, quality of life, depression, perseverative thinking, resistance to change. Perhaps earlier in the introduction in describing Eds, the core components and cognitive/affective risk factors.

8. Line 124 - 126 information should be in the method section.

8. 1 Line 62-66 is an awkward long sentence. Recommend rewording.

9. The authors are clear in their intention to include those at risk of developing an ED but do not provide what criteria indicate this population of individuals. The reader infers this by reviewing closely the study characteristic table. Preferably the authors will include this information in their method section at the very least, and preferably also in the introduction setting up the rational for including the population of interest.

10. Line 160 is a typo in "were involved participants"

11. Line 167, please provide a citation for GRADE

12. Inconsistent use of COVID vs. COVID-19. Be consistent.

13. Line 366 the authors state that consistent with their findings, existing research shows that higher effectiveness, retention and satisfaction with guided programs vs unguided. While this is true with the overall research base it was not the focus of the reviewed paper therefore the presented paper is unable to say that their findings are consistent.

14. Similar to the organization of the introduction, I found the discussion section to be scattered, unorganized and at times repetitive. I encourage a thorough edit with an eye toward concise and clear writing.

6. PLOS authors have the option to publish the peer review history of their article (what does this mean? ). If published, this will include your full peer review and any attached files.

**Do you want your identity to be public for this peer review?** For information about this choice, including consent withdrawal, please see our Privacy Policy .

Reviewer #1: No

Reviewer #2: No

---

## [Decision Letter · Decision Letter 1]

18 Mar 2025

Effectiveness of unguided internet-based computer self-help platforms for eating disorders (with or without an associated app): A systematic review

PDIG-D-24-00478R1

Dear Miss Gentile,

We are pleased to inform you that your manuscript 'Effectiveness of unguided internet-based computer self-help platforms for eating disorders (with or without an associated app): A systematic review' has been provisionally accepted for publication in PLOS Digital Health.

Best regards,

Calvin Or, PhD

Section Editor

PLOS Digital Health

**Additional Editor Comments (if provided):**

**Reviewer Comments (if any, and for reference):**

Reviewer's Responses to Questions

**Comments to the Author**

1. If the authors have adequately addressed your comments raised in a previous round of review and you feel that this manuscript is now acceptable for publication, you may indicate that here to bypass the “Comments to the Author” section, enter your conflict of interest statement in the “Confidential to Editor” section, and submit your "Accept" recommendation.

Reviewer #1: All comments have been addressed

Reviewer #2: All comments have been addressed

2. Does this manuscript meet PLOS Digital Health’s publication criteria ? Is the manuscript technically sound, and do the data support the conclusions? The manuscript must describe methodologically and ethically rigorous research with conclusions that are appropriately drawn based on the data presented.

Reviewer #1: Yes

Reviewer #2: Yes

3. Has the statistical analysis been performed appropriately and rigorously?

Reviewer #1: Yes

Reviewer #2: N/A

4. Have the authors made all data underlying the findings in their manuscript fully available (please refer to the Data Availability Statement at the start of the manuscript PDF file)?

Reviewer #1: Yes

Reviewer #2: Yes

5. Is the manuscript presented in an intelligible fashion and written in standard English?

Reviewer #1: Yes

Reviewer #2: Yes

6. Review Comments to the Author

Reviewer #1: (No Response)

Reviewer #2: Thank you for the close attention paid to reviewer comments. The result is an improved document.

7. PLOS authors have the option to publish the peer review history of their article (what does this mean? ). If published, this will include your full peer review and any attached files.

**Do you want your identity to be public for this peer review?** For information about this choice, including consent withdrawal, please see our Privacy Policy .

Reviewer #1: **Yes: ** Kalee Lodewyk

Reviewer #2: No
